# Assessment of GNSS Galileo Contribution to the Modernization of CROPOS’s Services

**DOI:** 10.3390/s23052466

**Published:** 2023-02-23

**Authors:** Danijel Šugar, Ana Kliman, Željko Bačić, Zvonimir Nevistić

**Affiliations:** 1Faculty of Geodesy, University of Zagreb, Kačićeva 26, 10000 Zagreb, Croatia; 2German Aerospace Center (DLR), Institute of Communication and Navigation, Muenchener Str. 20, Oberpfaffenhofen, 82234 Wessling, Germany

**Keywords:** CROPOS, GNSS, Galileo, modernization, VPPS, GPPS

## Abstract

CROPOS, as the Croatian GNSS network, was modernized and upgraded to support the Galileo system in 2019. Two of CROPOS’s services—VPPS (Network RTK service) and GPPS (post-processing service)—were assessed for the contribution of the Galileo system to their performance. A station used for field testing was previously examined and surveyed to determine the local horizon and to carry out a detailed mission planning. The whole day of observation was divided into several sessions, each with a different visibility of Galileo satellites. A special observation sequence was designed: VPPS (GPS-GLO-GAL), VPPS (GAL-only), and GPPS (GPS-GLO-GAL-BDS). All observations were taken on the same station with the same GNSS receiver, Trimble R12. Each static observation session was post-processed in Trimble Business Center (TBC) in two different ways: considering all available systems (GGGB) and considering GAL-only observations. A daily static solution based on all systems (GGGB) was considered as the reference for the accuracy assessment of all obtained solutions. The results obtained with VPPS (GPS-GLO-GAL) and VPPS (GAL-only) were analyzed and assessed; the results obtained with GAL-only have shown a slightly higher scatter. It was concluded that the inclusion of the Galileo system in CROPOS has contributed to the availability and reliability of solutions but not to their accuracy. By complying with the observation rules and taking redundant measurements, the accuracy of GAL-only results can be improved.

## 1. Introduction

The single-base Real-Time Kinematic (RTK) GPS (GNSS) is a differential positioning method based on a double differenced carrier phase algorithm which relies on the On-The-Fly (OTF) ambiguity resolution method providing results with cm-level accuracy in real time [1]. The RTK method involves a reference receiver transmitting its raw measurements or observation corrections to a rover receiver via some sort of data communication link. The data processing at the rover site includes ambiguity resolution of the differenced carrier phase data and coordinates estimation of the rover position [2] with the accuracy of 1 cm + 1 ppm over a short range [3]. More specifically, Feng and Wang [4] have reported that the RTK horizontal and vertical accuracy can be specified versus the rover-base distances as 1 cm + 0.5 ppm and 2 cm + 1 ppm, respectively. The basic concept of a single-base RTK is to use real-time corrections from a reference station to reduce and remove errors common to a base and rover pair to achieve centimeter-level accuracy. The system concept was developed in the late 1980s and early 1990s by the academia, science, private, and government sectors to improve the efficiency, accuracy, and timeliness of dredging and hydrographic surveying. In 1993, the Trimble company released its first commercial RTK product. A year later, it released the first commercial RTK receiver with OTF capabilities. That opened the door to the utilization of RTK by surveyors and geospatial professionals. A comprehensive history of the RTK was portrayed by Hartmann [5,6,7,8]. Although RTK provided a substantial benefit to position determination, the weakness of the method soon became evident: the method could be applied up to distances of 10–20 km from the base receiver because of the spatial decorrelation of distance-dependent errors induced by the ionosphere, troposphere, and orbital errors [9,10]. Distance-dependent errors can be separated into a dispersive component, consisting mainly of ionospheric refraction, and a non-dispersive component, consisting of tropospheric refraction and orbit errors [10]. The ionosphere is subject to rapid and localized disturbances, and as such, it represents the main restriction on the reference station density, whereas the tropospheric and orbit errors are known to change only slowly with time [2,11].

To mitigate those weaknesses of the single-base RTK, the Network RTK (NRTK) was developed. It provides larger service area coverage, increased robustness, and higher positioning accuracy due to more precise accounting for spatially auto-correlated errors in GNSS measurements [12,13]. Compared with the single-base RTK, the advantage of network RTK is that large portions of ionospheric and geometric errors are removed through network corrections, enabling a considerable improvement in RTK initialization and positioning accuracy [14]. Continuously Operating Reference Stations (CORS) network density is usually restricted to 70–100 km to allow for quick and reliable ambiguity resolution [3,10]. Over the years, several concepts of NRTK have been developed [15], the Virtual Base Station (VRS) concept being among the most used ones [2,10,16]. The VRS concept was announced in 2000 by Spectra Precision Terrasat company [8,17]. The VRS concept was outlined in, e.g., Landau et al. [18], its concept and performance in Janssen [15], the VRS observation algorithm was given in, e.g., Wei et al. [19], whereas the VRS data generation approach was given in, e.g., Hu et al. [12]. An overview of the development of the VRS is portrayed in Schrock [16]. The VRS requires a two-way communication channel between rover and server, where the corrections are streamed via the NTRIP (Networked Transport of RTCM via Internet Protocol). The GNSS errors are usually reduced via the Observation Space Representation (OSR) modelling method, which provides a single compound of ranging corrections as observed in a nearby (real or virtual) reference station [3]. Data processing at the system’s side uses the mathematically optimal Kalman filter technique to process data (observations) from reference stations in the network, in that way modelling all relevant error sources [20]. Once the network errors are computed at the reference station, distance-dependent errors need to be interpolated at the location of the user receiver [10]. The main purpose of the VRS station is to reduce the baseline distance between the rover and the reference station to facilitate the efficient removal of the spatially correlated errors using differential processing and to incorporate error corrections obtained from the network [21]. One of the core issues of NRTK is how to interpolate the distance-dependent biases generated from the network to the user’s (rover’s) location. Over the years, several methods were proposed for the interpolation of the distance-dependent residual biases [17]. The network solutions increase the reliability and productivity of ambiguity resolution and the positioning accuracy of rovers working in the system [22]. As outlined by Paziewski and Wielgosz [23], the overall procedure for NRTK positioning methodology consists of three steps: (1) the processing of the reference network GNSS data to derive the network corrections, (2) the interpolation of ionospheric and tropospheric corrections for the user location, and (3) user solution position determination with the application of the network-derived corrections. Current techniques (e.g., Network-RTK) are based on Local and Regional Networks of GNSS Reference Receivers, with inter-distances in the order of 70 km. Such networks are commonly managed by Land Administration or Geodetic Authorities [24]. That is the case with CROPOS (Croatian Positioning System)—a national permanent GNSS network in Croatia which was modernized and upgraded in 2019 to provide support for the European Galileo satellite system. The assessment of the Galileo satellite system’s contribution to the performance of CROPOS’s services is the core of this article.

## 2. Materials and Methods

### 2.1. CROPOS

CROPOS as a system was established in December of 2008. At that time, the network consisted of 30 stations, evenly distributed across the national territory at an average distance between stations of 70 km, and of a control center located in the headquarters of the State Geodetic Administration (SGA) in Zagreb. Later, the network on the national territory was enlarged by 3 stations, bringing the total number of CROPOS stations to 33. In order to improve the reliability of corrections in the areas near the border with neighboring countries, an additional 18 stations from Slovenia (7 stations), Bosnia and Herzegovina (5 stations), Hungary (4 stations), and Montenegro (2 stations) were included in the networked solution and correction parameters calculation, bringing the number of stations currently included in the networked solution to 51 [25]. The operation of the system relies on the VRS concept. The Trimble’s GNSS infrastructure installed in 2008 was providing support for GPS and GLONASS systems. Since its establishment, CROPOS has been widely accepted by the surveying community in Croatia, offering three services: DSP, VPPS, and GPPS. DSP service is based on the networked solution of code observations providing 0.5 m accuracy level, and it is used mainly for GIS applications. The VPPS service (network RTK) is the most used service of CROPOS, providing 2 cm for 2D accuracy and 4 cm for 3D accuracy [26]. The Geodetic Precise Positioning Service (GPPS) provides observation data in different RINEX and Trimble proprietary formats for post-processing. The data are available at CROPOS GNSS REFERENCE STATION WEB SERVER [27] to the subscribed users for 51 CORS stations or arbitrary VRS stations with the following logging rates: 1, 2, 5, 10, 15, 20, 30, and 60 s. The modernization of CROPOS started in 2018, and it was finished by the end of August 2019. Within the modernization project, all existing installed GNSS receivers and antennas were replaced by the latest Trimble GNSS receiver Alloy and the accompanying GNSS antenna Zephyr Geodetic III [28]. The control center software was upgraded to Trimble Pivot Platform 4.1, enabling the correction determination and service delivery based on GPS, GLONASS (GLO), and Galileo (GAL) systems. Several mount points were added to the VPPS service (e.g., CROPOS_VRS_GGG_RTCM32), thus providing support for the Galileo system in RTCM 3.2 MSM format (Radio Technical Commission for Maritime (RTCM) Multiple Signal Messages (MSM)) [29]. Since the modernization, the GPPS service has provided observation data for all GNSSs in various RINEX format versions (2.10, 2.11, 3.02, 3.03, 3.04) as well as in Trimble proprietary formats (T01, T02). According to the latest legislation (Law on Amendments to the Law on State Survey and Real Estate Cadastre), since April of 2022, the DSP and VPPS services are free of charge [30].

### 2.2. Galileo Satellite System and Its Applicability for High-Accuracy Positioning

Galileo is the European Global Navigation Satellite System (EGNSS), under civil control, providing PVT (Position, Velocity, and Time) service to users worldwide. The whole program is fully funded and owned by the European Union (EU). It was designed and developed by ESA (European Space Agency) and is operated by EUSPA (European Union Agency for the Space Programme) [31]. Currently, the Galileo system is under development. A comprehensive overview of the early stages of Galileo system development is depicted in Bonnor [32]; an overview of the Galileo Full Operational Capability (FOC) is supplied on the ESA Earth Observation Portal [33]. The information about the current constellation is available on the web site of the European GNSS Service Centre [34].

Galileo transmits several signals and codes on four different carrier frequencies within the range of 1.1–1.6 GHz, namely: E1 (1575.42 MHz; observation codes: C1A, C1B, C1C, C1X, C1Z), E5a (1176.45 MHz; observation codes: C5I, C5Q, C5X), E5b (1207.14 MHz; observation codes: C7I, C7Q, C7X), and E6 (1278.75 MHz; observation codes: C6A, C6B, C6C, C6X, C6Z) and multiplexed together through the AltBOC (Alternative Binary Offset Carrier) scheme [3,35,36]. The AltBOC signal, as the unique feature of Galileo in the combined E5a + E5b (known as E5, 1191.795 MHz; observation codes: C8I, C8Q, C8X) band, offers superior noise and multipath performance [37]. Moreover, the wideband E5 signal is expected to have a much better code (pseudorange) accuracy as compared to its subcarriers E5a and E5b [38]. The bandwidth of E5 is 51.15 MHz, which is far larger than the other signals, and this may explain the superior performance of signal E5 [39].

The very first autonomous determination of ground coordinates using the four Galileo satellites took place in March 2013, marking a milestone in Galileo development [40]. Before that, in January 2013, with four Galileo IOV (In Orbit Validation) satellites in constellation, the short period of their visibility allowed the demonstration of absolute and relative positioning using measurements from GAL satellites only. External orbit and clock information was necessary, since the IOV satellites were not transmitting a valid navigation message at that time. The Galileo frequencies E1 and E5a were observed for absolute positioning, whereas the relative positionings were computed from Galileo E1, E5a, E5b, and E5 AltBOC carrier-phase observations. The test has shown that E1 and E5 carrier-phase measurements will be particularly relevant for future positioning applications due to the possibility of mixed-constellation ambiguity resolution with GPS L1 and L5 signals [41].

The results of RTK positioning based on carrier-phase and code (pseudorange) observations of four Galileo IOV satellites broadcasting navigation message data were presented in [38]. The results of Galileo IOV ambiguity resolution, standalone as well as with GPS, were described for the very first time. The results have demonstrated that the integer ambiguity resolution based on the four IOV satellites needs fewer than three minutes when observables from at least three frequencies are used. Combined with data of four GPS satellites, even instantaneous (single epoch) ambiguity resolution is demonstrated using only two frequencies per constellation (i.e., E1 + E5a and L1 + L2).

The Galileo E1 and E5a bands are aligned to GPS L1 and L5, respectively. The advantage of multi-frequency carrier phase measurement is the diversity of carrier wavelengths helping to isolate the correct integer carrier phase ambiguities quickly and reliably. Dual-frequency GNSS observations are sufficient for ionospheric bias estimation/compensation and robust integer carrier phase ambiguity resolution. The addition of carrier phase and code measurements on a third frequency further strengthens ionospheric estimation and the speed and reliability of carrier phase ambiguity resolution [42].

The Galileo system, once fully deployed, will offer five high performance services worldwide: Open Service (OS), Public Regulated Service (PRS), High Accuracy Service (HAS), Commercial Authentication Service (CAS), and Search and Rescue Service (SAR) [43]. The Galileo OS is offered free of charge to all users, providing ranging positioning and timing service in three different frequency bands, enabling single (SF)- and dual (DF)- frequency positioning for users equipped with suitable receivers. Galileo’s free-of-charge Open Service relying on SF or DF improves the augmentation services as RTK/DGNSS or PPP (Precise Point Positioning). The resulting benefits to surveyors, especially in multi-constellation environments, include: easier mitigation of multipath errors, higher signal-to-noise ratio (SNR), increased availability, continuity and reliability, and better operation in such harsh environments as urban/natural canyons or under tree canopies [44].

The feasibility of Galileo-only observations for single-base RTK positioning was assessed by Mihoković et al. [45]. It was pointed out that due to the small number of available satellites and constellation under construction, the Galileo system was not ready for reliable individual RTK positioning. With the visibility of five or more Galileo satellites, the fixed solution was feasible, the deviations from the ground truth (reference) solution have shown value beyond the expected 1-cm level (max. 43 mm of horizontal deviation was recorded), and that level was obtained when all systems (GGGB = GPS + GLO + GAL + BDS) were used. Interestingly, when Galileo observations were combined with at least one FOC system (GPS or GLONASS), the results with deviation up to 2 cm were obtained (horizontally and vertically).

The contribution of the modernization of GPS and the introduction of Galileo to the positioning performance was assessed by Chen et al. [22]: compared to dual-frequency RTK, one of the main advantages of the third/fourth frequencies is the dramatic increase in the reliability and productivity of OTF initializations. The errors which are not frequency-dependent (e.g., troposphere and orbit) will not be removed by adding more frequencies: they will contribute to the multipath mitigation due to the availability of more observables. The addition of more frequencies will be beneficial to the Network RTK, enabling the increase in the interstation spacing because the ambiguity resolution is more robust against the ionosphere when more carriers are used.

The performance of two and three carriers in RTK positioning in various single-base and network scenarios was demonstrated in Vollath et al. [14]. The software simulations for the generation of code and carrier data of GPS and Galileo satellites have shown that in the presence of a reference station network, RTK initialization and positioning accuracy are improved considerably. The use of a third frequency and of the GPS and GAL satellite constellation has shown an increase in reliability for single baselines as well as for VRS solutions. The third frequency makes the ambiguity resolution more robust in difficult ionospheric conditions which arise from interpolation errors in larger networks.

The benefits of Galileo’s contribution for high-precision RTK have been shown through four comprehensive case studies, considering different baseline lengths, multipath environment, and tree canopy in [46]. The results have confirmed the usability of the current constellation (April 2017) in RTK applications and have shown the improvement in availability, reliability, and time-to-fix parameters. One case study addressed the Galileo-only RTK. Due to the limited number of usable GAL satellites, GAL-only RTK positioning was carried out in open-sky conditions over a very short baseline of 1 m. A single-base RTK positioning was performed with four-system corrections where the accuracy was assessed from GPS-only, GLO-only, and GAL-only solutions. The analysis has shown that 3D errors from GPS-only and GAL-only were at a comparable level.

An assessment of the benefits of Galileo to high-precision GNSS positioning (RTK, PPP, Post-Processing) was tackled in Luo et al. [47]. The post-processing studies have shown that the inclusion of Galileo provided a more accurate solution for short data durations and long baselines, particularly for data collected under strong multipath conditions. The feasibility of a post-processing solution based on GAL-only was shown in Šugar et al. [48], where the comparison of solutions obtained from different GNSS combinations was presented, showing the potential of individual and combined solutions. Paziewski and Wielgosz [23] have confirmed that the application of multi-frequency observations in the case of the Galileo system has an advantage over a dual-frequency solution. The key factor in relative positioning is the resolution of double-differenced ambiguities. The application of more than two frequencies can be beneficial for ionosphere modeling, which is crucial for the ambiguity resolution. Two overlapping frequencies (L1/GPS and E1/Galileo, and L5/GPS and E5a/Galileo) will allow double-differenced observations to be created between both systems.

The performance of the new Galileo (GAL) and BeiDou (BDS) satellites and signals was evaluated in Tian et al. [39]. These signals were assessed in regard to the carrier-to-noise density ratio (C/N0), code multipath (MP) combination, and triple-frequency carrier phase ionospheric-free and geometry-free (DIF) combination. Although there were no significant differences for the RTK results of the three systems, the double-differenced carrier phase and code residuals of E5 are the smallest among all signals.

The situation and performance of the five nationwide and local permanent GNSS networks in Poland featuring over 500 stations and providing access to NRTK services was assessed by Prochniewicz et al. [49]. A comprehensive test was conducted to evaluate the quality of network-based GNSS positioning services’ performance provided by all CORS networks available in Poland. It was concluded that no significant differences in positioning results were observed for different combinations of GNSS systems. In particular, the use of four GNSS systems did not lead to an evident improvement in positioning accuracy, but a small increase in positioning accuracy was noticeable as the number of GNSS systems used increased. SWEPOS is the CORS network for satellite positioning in Sweden. Since the introduction of Galileo to its service in 2018, the network users have observed higher availability and better performances, especially when using a high cut-off angle or in harsh environments [50]. An up-to-date overview of the development and the current status of SWEPOS was described by Schrock [51]. The very first results of the combined Galileo and GPS RTK solution in 2016 have shown an increase in the number of fixed solutions and a shortening of the times required for a fixed solution. Now that surveyors all around the world are able to use the Galileo signal for positioning following the declaration of Initial Services in 2016, more than 50% providers of the NRTK services have already upgraded or have started to upgrade to Galileo. The field tests have shown the benefits of the inclusion of Galileo into the RTK networks: improved availability, reliability, accuracy, and time-to-fix in difficult measuring environments such as urban canyons and under tree canopies [52]. The majority of RTK providers have upgraded or have started to upgrade to the Galileo system: SWEPOS (SE), GeoSoft (ET), SAPOS (DE), SOGEI (IT), GEONET (JP), TERIA (FR), etc. [53].

### 2.3. Mission Planning

Since the Galileo constellation is still not complete, mission planning was needed to identify and assess the visibility of Galileo satellites. For assessing the visibility of GNSS satellites along with the estimation of PDOP (Position Dilution of Precision) values, two tools were used, both provided by the Trimble company: Planning (version 2.90) and the Trimble GNSS Planning Online, available on Trimble [54]. These tools can be regarded as mutually complementary since certain functionalities present in one tool are not present in the other one. For both tools, the almanac data are essential: for the offline tool, the data have to be downloaded in the appropriate file format (e.g., alm), whereas for the online tool, the almanac data are automatically retrieved. The advantage of the offline tool is that a real horizon can be defined, the duration of the time window can start at any time of the day, and all quantities can be calculated (e.g., various DOP values, elevation, azimuth, and number of visible satellites broken down by each individual system (GPS, GLO, GAL, BDS)). The drawback of the online tool is that it enables the duration of the time window for 6, 12, or 24 h starting at hour (01:00, 02:00, etc.), but on the other hand, it offers the information about TEC (Total Electron Content), Ionospheric index, and scintillation. The visible (or local) horizon around the station was surveyed using a total station; subsequently, the azimuth and elevation values of the visible horizon were derived and finally imported into the Planning offline tool. The local or visible horizon is the ‘line where Earth and sky seem to meet but are blocked by elevated features of the landscape, such as trees or mountains’ [55]. The sky plot, produced in the Planning tool and showing the local horizon and Galileo satellites, is shown in Figure 1.

A great advantage of the offline tool is that different PDOPs (GDOP, PDOP, HDOP, VDOP, TDOP) along with the number of visible satellites above the elevation mask can be calculated and displayed for a 1-min interval or longer. Since the field testing was focused on the assessment of Galileo’s contribution to the performance of CROPOS’s services, the visibility of Galileo satellites was analyzed. At the time of field GNSS observations (July 2020), GPS and GLONASS systems had a declared Full Operational Capability whereas the Galileo system was under construction, with activities ongoing towards the declaration of FOC. The FOC of the BeiDou system (BDS-3) was declared only two weeks after the conducted measurements, on 31 July 2020 [56]. Focusing on the Galileo satellites’ almanac data only, the planning was carried out for the day 14 July 2020 (DOY 196) starting at 8:00 through 17:00 Central European Time (CET). Several time windows were considered with different DOP values and number of visible GAL satellites. Upon analysis, nine time-windows were identified with good, medium, and bad visibility. At the end of the day, a short time window was identified with only four GAL satellites visible, which should prevent the availability of a fixed RTK solution (for OTF ambiguity resolution, the minimum visibility of five satellites is needed).

According to the number of visible GAL satellites at the station A above the visible horizon for the time window 8–17 CET, an observation plan was set up and is presented in Table 1.

The times (epochs) highlighted in Table 1 are indicated in Figure 2, where all DOP values calculated upon Galileo satellites’ visibility are displayed. The Galileo constellation consisted of 22 operational satellites in July of 2020. In the time-window 6–18 CET, there were 4–8 visible satellites having PDOP values in the range of 1.67–4.29. The basic idea was to carry out the observations (kinematic and static) in 10 sessions with a different number of visible satellites (4–8) and consequently with different PDOP values (1.67–4.29) to gain insight into the performance of CROPOS’s services using GAL-only observations. Out of 10 sessions, 4 sessions were assumed to have good observation conditions (1, 4, 7, and 9), another 4 sessions were assumed to have bad observation conditions (2, 6, 8, and 10), whereas the remaining sessions 3 and 5 had medium observation conditions. Due to the short time foreseen for the session, after sessions 1 and 9, there were no observed static observations.

### 2.4. Preparation and Execution of the Field Activities

All GNSS measurements were carried out at station A located at the football playground in the vicinity of the Faculty of Geodesy in Zagreb, Croatia. The visible horizon with elevations below 18° was in the portion of sky with azimuths in the range of 45°–315°, where most signals of visible satellites come from (see Figure 1). In Figure 3, the visible horizon around the station A is displayed with a red line. The azimuth and elevation values of the visible horizon were determined with total station measurements.

To leverage the full potential of the modernized CROPOS and all GNSS satellites (GPS, GLO, GAL, BDS), a new GNSS receiver, Trimble R12, was used in July 2020. The receiver was introduced to the market in November 2019 [57]. It uses the Trimble ProPoint GNSS technology, allowing for a flexible signal management, helping to mitigate the effects of the signal degradation, and providing a GNSS constellation-agnostic operation. According to the DataSheet [58], the Trimble R12 receiver has 672 channels tracking all GNSS systems: GPS, GLO, GAL, BDS, QZSS, NAVIC, and SBAS. Regarding the Galileo system, all frequencies are supported: E1, E5a, E5b, E5 AltBOC, E6. Across all observation sessions, the receiver was setup on the surveying pole (fixed height 2.000 m measured up to the bottom of the quick release) and supported by a bipod. Along with the GNSS receiver, the T7 tablet running the Win 10 operating system and the Trimble Access software ver. 2019.11 were used, supporting all surveying workflow on the field.

According to Annex 3 of the regulations for the basic geodetic work performance [59] related to the usage of CROPOS, it is regulated (foreseen) that the coordinates of the complementary network stations must be determined by VPPS in two independent sessions, each session consisting of three consecutive series, each series consisting of thirty epochs (seconds). Each series is to be performed with independent initialization. According to the regulations, those two sessions must be spanned by a minimum of two hours to ensure independent satellite visibility and independent calculated fixed solutions, with an elevation mask of 10°–15°, a minimum number of 5 satellites used, and PDOP < 6 [60]. As foreseen by the regulations, the final coordinate values are calculated as the average of all individual results with the precision estimation provided in the form of standard deviation.

A similar approach has been prescribed by the user guidelines for the single base real-time GNSS positioning [61] where the coordinates are obtained from sessions staggered by three or four hours, consequently having different satellite geometry and multipath effects. A similar methodology was devised by Odolinski [62]: two observation sessions (station occupations) are recommended with a time separation of 20–45 min to diminish the time correlation effects which occur due to the atmospheric conditions (ionosphere, troposphere) and multipath in combination with a slowly changing satellite geometry.

For the assessment of the contribution of Galileo satellites to the performance of CROPOS VPPS and GPPS services, three individual survey styles were created as follows: two styles (DIPLOMSKI_2020_VPPS and DIPLOMSKI_2020_Galileo) related to the VPPS (kinematic) and one related to the GPPS (static) observations (DIPLOMSKI_2020_GPPS). The kinematic survey styles were created to leverage the full potential of CROPOS as a system and Trimble R12 as a GNSS receiver; both survey styles featured the corrections in RTCM 3.2 MSM format, once supporting GGG systems (GPS, GLO, GAL), and the second time supporting the Galileo-only system. For both survey styles, a 10° elevation mask was applied, with 30 epochs and PDOP mask value set to 6 (default). The survey style related to static observations featured all satellites (GGGB), elevation mask 10°, logging interval 5 s, and session duration 22 min. The duration of a conventional static session was determined according to the rules set forth in Hofmann-Wellenhof et al. [63] for the dual-frequency (L1 + L2) receivers: 20 min + 2 min/km of the baseline length. Although those rules present a conservative approach for session length determination, they were accepted to ensure a reliable baseline determination. In addition to the close (nearby) CORS ZAGR station, observation data for two additional VRS station were downloaded in T02 format and used in baseline processing. All those stations were evenly distributed around the station A at distance < 1 km leading to the session length of 22 min. The CORS ZAGR station and two VRS stations form an approximately equilateral triangle with station A being in its center. The parameters of the survey styles are outlined as follows:VPPS (GGG): GGG, 10°, 30 s, RTCM 3.2 MSM, PDOP 6VPPS (GAL-only): GAL, 10°, 30 s, RTCM 3.2 MSM, PDOP 6GPPS (GGGB): GGGB, 10°, logging interval 5 s, duration 22 min.

Bearing in mind the planned times for observation sessions, the observation schema (sequence) was set as follows: three separate sessions (with individual initialization; each lasting for 30 s) using GGG satellite systems, followed by three separate sessions (with individual initialization; each lasting for 30 s) using Galileo-only signals, and finishing the observation window with 22 min of static observations. The observation session sequence is summarized as follows:VPPS (GGG): 30 s,VPPS (GGG): 30 s,VPPS (GGG): 30 s,VPPS (GAL-only): 30 s,VPPS (GAL-only): 30 s,VPPS (GAL-only): 30 s,GPPS (GGGB): 22 min.

The overall duration of the observation sequence was approximately 27 min. All VPPS sessions were carried out within 4–6 min, providing almost stable (unchanged) observation conditions and satellites visibility. During the field surveying activities, certain problems were encountered related mostly to connecting the rover receiver to the base (fourth session) as well as freezing of the Trimble Access software (sixth session) (see Table 2). All those events have delayed and caused modifications of the planned observation schedule. During the ninth session, VPPS (GGG) observations were carried out as planned, and the measurements with GAL-only observations were taken for only two series (out of three planned). Most likely, the visibility of one out of five visible GAL satellites has fallen below the elevation mask, preventing the acquisition of a fixed solution.

All sessions are outlined in Table 2, each series with start times (CET) of observations carried out, along with the information about PDOP and number of satellites contributed to the solution.

By comparing the time schedule (Table 1) to the times of sessions as they were carried out (Table 2), it can be easily seen that the first five sessions were performed as planned, but problems that occurred during the sixth session caused the modifications of the further sessions.

## 3. Results

### 3.1. Static Observations and Processing in Trimble Business Center

During the survey, the GNSS receiver (antenna) was oriented towards the north to mitigate the variation in the antenna phase centers. For the GNSS receiver Trimble R12, the North Reference Point (NRP) is the Man–Machine Interface (MMI). The antenna calibration values assume that the NRP is properly oriented to true north [64].

The baseline processing was carried out in Trimble Business Center (TBC) ver. 5.3. TBC is a program package with a GPS processing engine introduced in 2005. Over time, a GNSS processing engine has evolved, and since ver. 3.5 (released in 2015), it supports the independent GNSS constellation solutions including BeiDou-only, GLONASS-only, and BeiDou + GLONASS-only combinations. Starting with the TBC, ver. 3.90, Galileo-only post-processing baseline solution was enabled, as well [48]. In version 4.0 (released in 2017), a modernized approach for static GNSS baseline processing was introduced: multiple processing modes are dynamically chosen according to the baseline length and duration of observation sessions [65]. Moreover, from ver. 4.00 on, the new engine also supports three frequency post-processing, mixed-signal post-processing, and provides support for the Galileo E5A, E5B, and E5 AltBOC signals, improving the accuracy and reliability of the processed baseline solution [66]. The baseline processing and the subsequent network adjustment were carried out within several individual projects. After the import of CORS, ZAGR, and VRS observation data in T02 format, the observation data collected with the Trimble R12 receiver were imported in T04 format. Both VRS stations along with the CORS ZAGR station were selected (defined) so as to form an approximate equilateral triangle, with station A being approximately in the center of said triangle.

In each project, the following settings were applied: coordinate system HTRS96/TM (the official coordinate system of Croatia), GRS80 ellipsoid, and Transverse Mercator (TM) projection with the central meridian 16°30′ and scale factor 0.9999. For baselines processing, the following parameters were set: elevation mask 10°, broadcast ephemeris used, and confidence level display 95%. Each static session was processed twice: once using all available satellite system observations (GPS + GLO + GAL + BDS) and the second time using GAL-only data. Four different project types were created according to the schema outlined in Figure 4: GGGB project providing nine solutions (each session provided one solution), ‘GAL-only’ project providing nine solutions (each session provided one solution), ‘GGGB (daily solution)’ project providing one solution, and ‘GAL-only (daily solution)’ providing one solution.

The network adjustment was performed in each project according to the procedure outlined in Trimble Geospatial [67]. It was carried out in two steps: minimally constrained network adjustment followed by the constrained network adjustment. In order to obtain two individual daily solutions, all sessions were combined, leading to a single solution based on GGGB observations and one solution based on GAL-only observations. Since the daily solution based on all available data (GGGB) was considered the most reliable one, it was considered as a REFERENCE solution for further analysis of the solutions obtained by kinematic VPPS observations. A single daily solution based on ‘GAL-only’ observations was compared with the REFERENCE solution. The differences ΔE = 0.0004 m, ΔN = −0.0001 m, and ΔH = −0.0010 m (displayed in Figure 5) have provided almost perfect matching between the two results. 

If each individual session result based on GAL-only data is compared to the REFERENCE, the following maximum absolute difference values are obtained: MAX (ABS (ΔE)) = 0.0046 m, MAX (ABS (ΔN)) = 0.0046 m, and MAX (ABS (ΔH)) = 0.0163 m. The differences between each individual ‘GAL-only’ static session and the REFERENCE are shown in Figure 6.

The values show a variable sign, confirming that the differences generally show stochastic (random behavior). Session A-6 presents an exception where the difference reached its maximum ΔH = 0.0163 m. When the 2D differences (planar differences between each solution and the REFERENCE) are examined, it turns out that the maximum value reached Δd = 0.0053 m, whereas the 3D (spatial) differences reached their maximum to ΔD = 0.0170 m. When the results of the A-6 session are excluded, the maximum value is ΔD = 0.0063 m. Overall, it can be stated that individual GAL-only sessions have enabled a reliable solution compared to the REFERENCE position obtained by all systems’ observation data. The random sign of coordinate differences across all sessions can be regarded as confirmation of the absence of significant biases. The analysis of observation data present in the RINEX observations files across all sessions led to the conclusion that the numbers of observed satellites were in the following intervals: GPS (7–10), GLO (5–7), GAL (5–8), and BDS (11–17), with all observed satellites being in the interval of 30–41. 

In the same manner, all individual sessions processed with all available observation data systems (GGGB) have shown slightly better results, with the maximum absolute value of coordinate differences given as follows: MAX (ABS (ΔE)) = 0.0030 m, MAX (ABS (ΔN)) = 0.0035 m, and MAX (ABS (ΔH)) = 0.0031 m leading to the MAX (Δd) = 4.4 mm and MAX (ΔD) = 4.6 mm. As expected, those results are better in comparison to results obtained with GAL-only individual sessions data. As in the previous analysis, all coordinate difference values show random (stochastic) behavior confirming the absence of significant biases. The differences between each individual static session and the REFERENCE are shown in Figure 7.

All individual planar (E, N) results obtained with ‘GAL-only’ and GGGB observation data are plotted together and displayed in Figure 8. All ‘GAL-only’ session results are within the circle with a radius of 6 mm, whereas all GGGB session results are within the circle having the radius of 5 mm. Interestingly, both sessions showing maximum departure from the REFERENCE belong to the same session: A-5.

### 3.2. Kinematic Observations Using VPPS

According to the observation schedule designed before the field measurement activities, each kinematic observation session consisted of three series (each series featuring 30 epochs). Before each series, an independent initialization was gained, meaning that the connection to the server was interrupted, subsequently re-established, and a new initialization was carried out, providing independent ambiguity fixed solutions. After three observation series using GPS-GLO-GAL (GGG) satellite systems, an additional three series using GAL-only observations were carried out. Upon the automatic connection to the mount point CROPOS_VRS_GGG_RTCM32, the correction data stream was established and, soon after, the fixed ambiguity solutions were obtained. The survey style regarding the coordinate determination with GAL-only data was slightly modified: the PDOP mask was set to the maximum value of 25 in order to make possible solutions in those circumstances when PDOP value would reach values above the default of 6 (e.g., ninth session with high PDOP values). Each solution was named according to the rule: ‘STATION-session #-series number within the session’ (e.g., A-1-1). The first three solutions (e.g., A-1-1, A-1-2, A-1-3) were obtained from GPS-GLO-GAL observations, the next three solutions were obtained with GAL-only observations (e.g., A-1-4, A-1-5, A-1-6), etc.

In almost all sessions, the VPPS measurements were performed in the time span of 4–6 min (see Table 2), providing very similar observation conditions and satellite visibility and thus enabling the comparison of the results. In that regard, the only exception was session #4, where a 30-min gap occurred between GGG and GAL-only observations owing to some problems with the connection to the base.

After the field activities, all created jobs were downloaded from the T7 device to the PC and converted to several ASCII files. Several file formats were combined to obtain the following information: Point # ID, Easting, Northing, Hz Precision, Vt Precision, PDOP, number of satellites, and time of observation.

A straightforward way to perform the analysis of the results obtained with the VPPS was to compare them with the REFERENCE. The values of the differences displayed in Figure 9 are in the range as follows: ΔE GGG (−4.3 mm to +5.2 mm) and ΔE GAL (−2.3 mm to 5.8 mm), confirming a slightly higher departure of the GAL solution from the REFERENCE. Interestingly, the RMS and STDEV values are given as follows: RMS ΔE (GGG) = 3.0 mm, STDEV ΔE (GGG) = 3.0 mm; RMS ΔE (GAL-only) = 3.2 mm, STDEV ΔE (GAL-only) = 2.9 mm.

When the ΔN differences are analyzed across all VPPS solutions, the results are given as follows. The values of the differences displayed in Figure 10 are in the range of ΔN GGG (−4.5 mm to +3.1 mm) and ΔN GAL-only (−6.5 mm to 4.9 mm). The range values (7.6 mm vs. 11.4 mm) along with the extreme value are higher for GAL-only solutions; the RMS (2.2 mm vs. 2.7 mm) and STDEV (2.1 mm vs. 2.6 mm) follow the same pattern, giving slightly higher values for GAL-only results. 

A similar analysis can be performed for the height component (Figure 11). The extreme values of ΔH are given in the range of GGG (−11.9 mm to +5.1 mm), GAL (−14.9 mm to +6.8 mm). The RMS and STDEV values are given as follows: GGG (RMS = 4.3 mm, STDEV = 3.4 mm), GAL-only (RMS = 6.3 mm, STDEV = 5.2 mm). This is the confirmation that GAL-only solutions have shown a higher deviation from the REFERENCE. That especially came into evidence during the last session (A-9), with six to seven satellites being initially available, and then the visibility dropped below five satellites, preventing the fixed solution. The differences show variable sign, mostly negative, confirming the presence of a certain bias between the VPPS and REFERENCE solution.

If the planar (2D) departure of VPPS (GGG) and VPPS (GAL-only) solutions from the REFERENCE position are considered, the following values are obtained: GGG (0.4 mm to 6.0 mm), GAL-only (+0.6 to +6.9 mm). This is an additional confirmation that MEDIAN (3.3 mm) and AVERAGE (3.5 mm) planar deviations of the VPPS (GGG) solutions are slightly smaller compared to GAL-only solutions: MEDIAN (4.0 mm) and AVERAGE (3.9 mm). If all solutions (27 GGG and 26 GAL-only) are plotted together, Figure 12 is displayed.

Compared to the REFERENCE solution, all VPPS (GGG) solutions are within the circle with the radius of 6 mm, whereas all VPPS (GAL-only) solutions are within the circle with radius of 7 mm.

## 4. Discussion

### 4.1. Analysis of the Number of Satellites That Contributed to VPPS Results and Estimated Precision Assessment

Considering all observations of VPPS series, it is worth analyzing the number of satellites which contributed to the solution along with the PDOP values and estimated Hz and Vt Precision (information obtained from the Trimble Access software, also available in real time during the survey). The range of values (MIN, MAX) is given in Table 3.

It can be clearly seen that the precision estimation is better for GGG than for GAL-only solutions. The AVERAGE and MEDIAN values of ‘Hz Precision’ for GGG solutions are 3.8 mm, whereas the AVERAGE and MEDIAN values of ‘Vt Precision’ for GGG solutions are 7.1 mm and 6.5 mm, respectively. When GAL-only solutions are considered, the AVERAGE and MEDIAN values are given as: 5.9 mm (AVERAGE) and 5.2 mm (MEDIAN) for Hz Precision and 9.9 mm (AVERAGE) and 8.7 mm (MEDIAN) for Vt Precision. Obviously, AVERAGE and MEDIAN values are almost equal for precision estimation based on ‘GGG’, whereas for ‘GAL-only’, there is a small difference (AVERAGE vs. MEDIAN) owing to a weaker (less populated) Galileo constellation.

Comparing the number of satellites that contributed to the solution with the number of visible satellites during the static session (22 min), it came into evidence that no BDS satellites contributed to the VPPS solutions.

Horizontal precision estimation (Hz Precision) in Trimble Access software was expressed as DRMS (Distance Root Mean Square), whereas the vertical precision estimation (Vt Precision) was given in terms of 1σ. DRMS is an estimate of the root mean square of the radial distance between the true position and the observed position [68]. According to the parameters for precision display in Trimble Access software, the Hz Precision is expressed as DRMS (63.2% probability that the estimate is within the range), the Vt Precision is given by 1σ with the probability of 68.3% that the height estimate is within the range from the true value [69,70].

Horizontal deviation of each VPPS solution from the REFERENCE value (Δd) across all sessions was compared to the ‘Hz Precision’ estimates given by Trimble Access software. Regarding the VPPS (GAL-only) solutions, the differences ‘Hz Precision–Δd’ are in the range (−0.9 mm to +7.5 mm), mostly positive (23/26 = 88%), whereas the differences ‘Vt Precision–ABS (ΔH)’ are in the range (−3.5 mm to +15.7 mm), mostly positive (22/26 = 85%), which means that the precision estimates given by Trimble Access software are pretty reliable (in approx. 90% of cases).

Regarding the same analysis for the solution obtained with GGG satellites data, the situation is as follows: the differences ‘Hz Precision–Δd’ are in the range (−2.5 mm to +3.4 mm), mostly positive (16/27 = 59%), whereas the differences ‘Vt Precision–ABS (ΔH)’ are in the range (−0.7 mm to +7.7 mm), mostly positive (26/27 = 96%), which means that the Vt. precision estimates are even more reliable. ‘Hz Precision’ estimates for VPPS (GAL-only) solutions are on average 2 mm higher (GGG (3.8 mm) vs. GAL-only (5.8 mm), and ‘Vt Precision’ estimates for VPPS (GAL-only) solutions are on average almost 3 mm higher (GGG (7.1 mm) vs. GAL-only (9.9 mm)). It can be concluded that for GAL-only solutions, the precision estimates for Horizontal and Vertical precision are rather conservative; for the GGG solution, they are more realistic. Nevertheless, in both cases, the Hz and Vt Precision estimates can be regarded as reliable. Across all sessions, the average PDOP value for GGG was 1.325, for GAL-only, it was 2.760, the average of visible GGG satellites was 20, and the average of visible GAL satellites was 6. The improved availability of GGG satellites has led to a better precision estimation compared to GAL-only solutions.

By comparing the number of satellites included in the observation RINEX 3.03 files, the solutions obtained with VPPS (GGG), and VPPS (GAL-only), it was confirmed that VPPS (GGG) solutions were gained by the contribution of GPS, GLO, and GAL satellites, whereas the BDS satellites have not contributed to the VPPS solution.

Moreover, the number of satellites in the first several recorded epochs in RINEX 3.03 observation files were compared to the closest VPPS solutions based on GGG and GAL-only observations. The analysis is given for all sessions containing static occupations. The results are summarized in Table 4, where the times are given for the start of each observation series (static, VPPS (GGG), VPPS (GAL-only)). The last session (ninth session) was not taken into consideration since there was a large time span between the instant of VPPS (GGG) and VPPS (GAL-only) observations and the start of the static session, and therefore, such a comparison would not be meaningful.

From the data given in Table 4, it can be inferred that there is a small difference (0–1) between the number of GAL satellites included in the VPPS (GAL-only) solutions and GAL satellites recorded in the RINEX observation file. That difference (which is always in favor of a static observation file) can be attributed to the fact that the visible GAL observations are recorded in the RINEX files whereas, in the VPPS solutions, only those GAL satellites simultaneously visible by the network and the receiver were contained. The time difference between the epochs of observation files and VPPS solutions is in the range 4–33 min, but those time differences have not caused a significant difference in the number of GAL satellites contained in the solutions, owing to the slowly changing visibility of GAL satellites.

A similar conclusion can be drawn considering the number of GGG satellites contributing to the VPPS (GGG) solution and the number of recorded GGG observations in the RINEX observation file: although the numbers of SV are similar, the difference between the epochs of observation (4–33) has contributed to the difference in number of SV contained in the solution (0–4). Additionally, this can be regarded as a confirmation that the VPPS solution relies on the observations (corrections) of the GGG satellites.

### 4.2. Analysis and Discussion of the VPPS Results

In general, a box and whisker chart shows a distribution of data into quartiles, highlighting the mean and outliers [71]. The interquartile range (IQR) measures the spread of the middle half of the data [41]; it is the range for the middle 50% of the data sample [72]. The lowest quartile (Q1) covers the smallest quarter of values in the dataset, and the upper quartile (Q4) comprises the highest quarter of values. The interquartile range is the middle half of the data that lies between the upper and lower quartiles. In other words, the interquartile range includes the 50% of data points that are above Q1 and below Q4. Unlike the more familiar mean and standard deviation, the interquartile range and the median are more robust measures. By comparing the size of the boxes, the variability of the data in the dataset can be understood: more dispersed distributions have wider boxes.

A total number of 27 VPPS (GGG) results gained through 9 sessions (27 = 9 × 3) using GGG observations along with the almost simultaneous 26 VPPS (GAL) results gained through 9 sessions (26 = 9 × 3 − 1) have enabled the box and whisker plots of Easting, Northing, and Height components. The results obtained with VPPS (GGG) and VPPS (GAL-only) observations are compared with the REFERENCE (REF) and shown in Figure 13 (Easting, Northing, Height).

Analysis and interpretation of results shown in Figure 13 are provided for Easting, Northing, and Height components as follows:Easting (Figure 13, left). The range of the Easting results obtained by GGG and ‘GAL-only’ are 0.0095 m and 0.0081 m, respectively. The greater range for GGG is due to the results obtained in session A-6 having higher DOP values (see Figure 9). The IQR for GGG solution is a bit higher (0.0056 m) compared to the ‘GAL-only’ solution (0.0054 m). Interestingly, the median values are almost equal (0.2 mm of difference), whereas there is a small difference between the average values of GGG (0.0176 m) vs. ‘GAL-only’ (0.0184). Both median values compared to the REF value supply the differences of 0.0009 m and 0.0011 m (GGG vs. ‘GAL-only’), which can be considered as a negligible amount. The averaging of VPPS results provides a more accurate and reliable solution. Easting components obtained with GGG and GAL-only observations have shown a great level of similarity (matching), leading to the conclusion that Easting components obtained with ‘GAL-only’ observations can be regarded as equally accurate and reliable as those obtained with GGG observations.Northing (Figure 13 middle). There is a small difference between median values: GGG (0.0276 m) vs. GAL-only (0.0277 m). There is a small difference between the average values: GGG (0.0273 m) vs. GAL-only (0.0270 m). The IQR is slightly smaller for GGG (0.0039 m) vs. GAL-only (0.0041 m), but the range for GAL-only is bigger (0.0114 m) compared to GGG (0.0076 m). That means that there is a higher spread of the results obtained with GAL-only observations, which can be seen in STDEV values, as well (0.0021 m vs. 0.0026 m). The higher range for Northing components obtained with GAL-only comes from the results in session A-8 (see Figure 10). Northing components obtained with GGG and GAL-only observations have shown a great level of similarity (matching), leading to the conclusion that Northing components obtained with ‘GAL-only’ observation can be regarded as equally accurate and reliable as those obtained with GGG observations, even though ‘GAL-only’ results have shown a greater level of variability. When the results are averaged and compared to the REFERENCE value, there is a negligible difference < 1 mm.Height (Figure 13 right). When it comes to the height components, there is no significant difference between MEDIAN values (GGG: 0.7806 m vs. GAL-only: 0.7799 m) and AVERAGE values (GGG: 0.7803 m vs GAL-only: 0.7794 m). On the other hand, there is a greater RANGE for ‘GAL-only’ results (0.0217 m) compared to GGG results (0.0171 m), that has consequences for the differences in IQR values (GGG: 0.0040 m vs. GAL-only: 0.0051 m). Small differences between the average values (<1 mm) compared to the REFERENCE value give the difference < 4 mm, providing a confirmation that calculating the average results improves the accuracy and reliability of the final value. Four outlier values have been found: one outlier in GGG results and the other three outliers in GAL-only results. Usually, the outliers are found by considering the IQR, Q1, and Q3 values, where the lower and upper outlier gates are calculated with the following expressions:
Q1 − 1.5 × IQR: Lower outlier gate.
Q3 + 1.5 × IQR: Upper outlier gate.

For the lower gate of GGG results, it applies that Q1 − 1.5 × IQR = 163.7782 − 1.5 × 0.0040 = 163.7722, which is greater than the value of 163.7711 which is, in its turn, identified as the outlier shown in Figure 13 (right). The height identified as an outlier was obtained as the first observation in the second series (A-2-1); comparing the subsequent results in the same series, those values have changed and came closer to the REFERENCE value (see Figure 11). That outlier could be related to some initialization uncertainties; subsequent series in the same session (A-2) with independent initializations have provided values closer to the REFERENCE. Lower and upper outlier gates for GAL-only heights are given as follows: Q1 − 1.5 × IQR = 163.7689 m, Q3 + 1.5 × IQR = 163.7892 m. When all GAL-only heights are compared with the lower and upper gate values, two outliers below the lower gate and one outlier above the upper gate are identified, as shown in Figure 13 (right). Two outliers below the lower gate came from the series A-9-1 and A-9-2 in the last session; the outlier above the upper gate came from the series A-1-3 in the first session (see Figure 13 right). Two outliers below the lower gate are related to the high DOP values and low visibility of Galileo satellites (Table 2). It can be concluded that GAL-only results have provided accurate and reliable results, although with a greater range and spread, pointing out that several sessions providing redundant measurements are needed to obtain more accurate and reliable results.

If all VPPS results (GGG and GAL-only) along with the GPPS (GGGB) are considered, and if the MIN and MAX values are analyzed, then the range values can be derived and displayed in Table 5.

Considering the range values, it can be clearly seen that the GPPS results are generally better than the VPPS results. Considering the VPPS results, a smaller range has been reached for solutions based on GGG than those based on GAL-only observations. A similar conclusion can be drawn for GPPS solutions: GGGB solutions show smaller range values than solutions based on GAL-only.

## 5. Conclusions

The modernization of CROPOS carried out in 2019 has enabled the inclusion of Galileo and BeiDou observation data into a networked solution. The performance of CROPOS prior the modernization was already at a high level; therefore, a significant improvement in terms of accuracy has not been achieved. The declared accuracy of CROPOS VPPS service of 2 cm (2D) and 4 cm (3D) has not been changed. The improvement in CROPOS’s modernization can be seen through the increased availability and reliability of solutions. The number of available satellites has increased from 54 (31 GPS + 23 GLO) to 126 (31 GPS + 23 GLO + 22 GAL + 50 BDS) (July 2020). Although the GAL system has not yet reached its FOC, when it is able to provide an independent VPPS solution (GAL-only), that solution can be obtained with the declared accuracy of CROPOS VPPS. Several facts (circumstances) must be pointed out to correctly assess the results: the test was performed leveraging the potential of a sophisticated and almost brand-new model of GNSS receiver (Trimble R12), the station had an almost free horizon with a weak or even absent source of multipath, and, equally important, the measurements were performed strictly according to the provisions of the regulations for the basic work performance. Considering the results of VPPS positioning based on GAL-only observations, the greatest deviation from the daily solution was 15 mm (vertical component), which was within the declared accuracy level. Since the Galileo constellation is still not fully deployed, there are time windows with less than four visible satellites preventing the kinematic fixed solution in real-time. Considering the individual static solutions based on GGGB and GAL-only observations, it has been shown that GAL-only solutions showed a slightly greater spread (6 mm) in comparison to a GGGB solution (5 mm). The same can be stated for the results obtained with VPPS (GGG) and VPPS (GAL-only) results: all GAL-only results are within the radius of 7 mm, and all GGG results are within the radius of 6 mm from the REFERENCE. It has been demonstrated that taking redundant measurements (observation sessions) and averaging the results lead to accuracy and reliability improvement. By the declaration of FOC in the times to come, it is expected that the Galileo constellation will provide additional benefits and robustness to CROPOS as a system and to the performance of its services, as well. It must be pointed out that the study presented in this article has been carried out using a Trimble GNSS receiver which was supposed to work the best within a CROPOS network relying on Trimble’s infrastructure. Further tests with GNSS receivers from other manufacturers will be carried out in the next period.

## Figures and Tables

**Figure 1 sensors-23-02466-f001:**
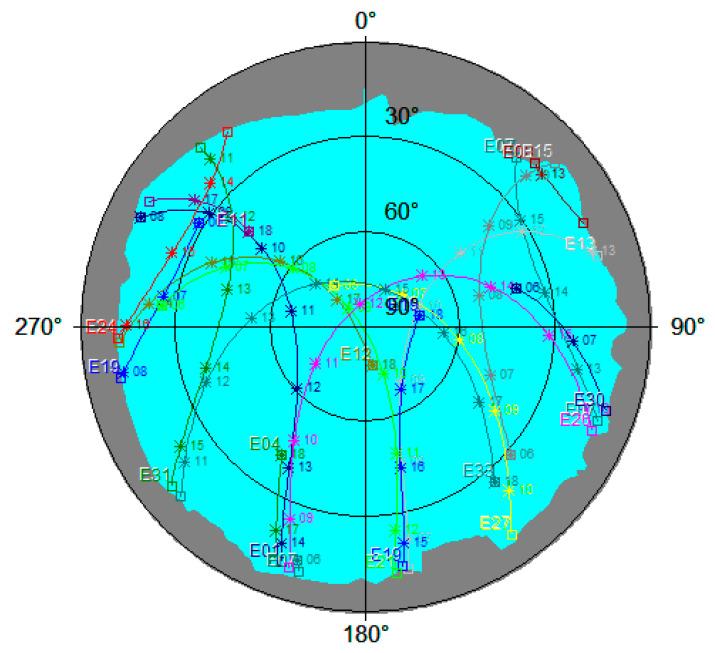
Sky Plot (Trimble Planning) of the station A (Zagreb, Croatia) for the time window 06:00–18:00 CET (14 July 2020) considering the local horizon and Galileo satellites only. The cyan area shows the open sky where the satellites’ signals come from. Each satellite’s track is marked by the GAL satellite it belongs to (e.g., E33); on each track are marked the positions planned for the full-hour epochs (e.g., 11, 12 … 18).

**Figure 2 sensors-23-02466-f002:**
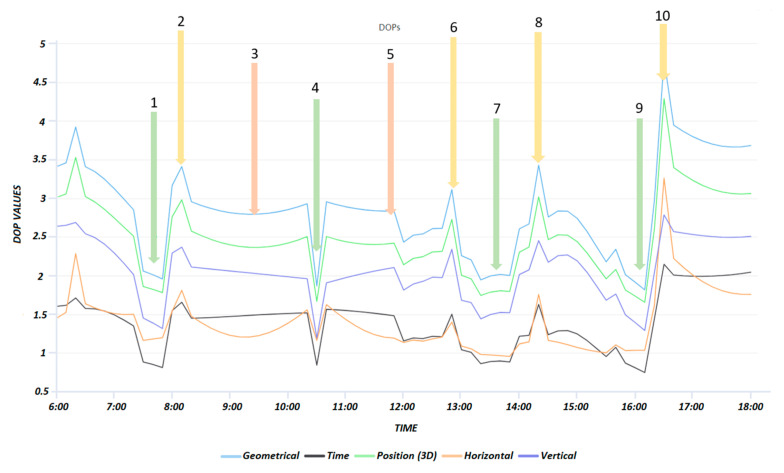
DOPs (Trimble GNSS Planning online) with times of planned observation sessions in the time window 06:00–18:00 CET (14 July 2020). The sessions planned are marked with numbers and arrows.

**Figure 3 sensors-23-02466-f003:**
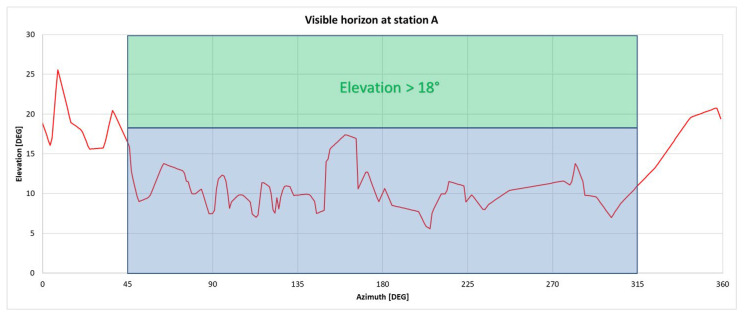
Visible horizon at station A (red line): maximum elevation (26°) of the visible horizon is in the azimuth 9°; in the portion of the sky 45° < Azimuth < 315°, the visible horizon is free above the elevation 18°.

**Figure 4 sensors-23-02466-f004:**
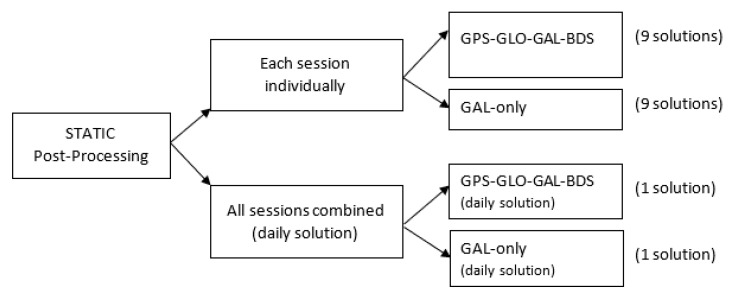
Static post-processing schema in TBC: each session was processed providing solutions based on GGGB and GAL-only observation data.

**Figure 5 sensors-23-02466-f005:**
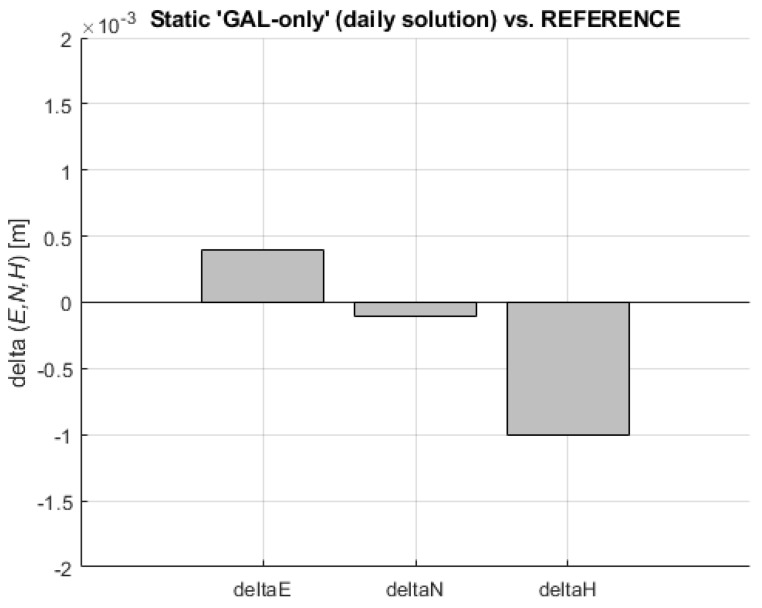
Coordinate differences (delta E, delta N, delta H) of static ‘GAL-only’ (daily solution)–REFERENCE.

**Figure 6 sensors-23-02466-f006:**
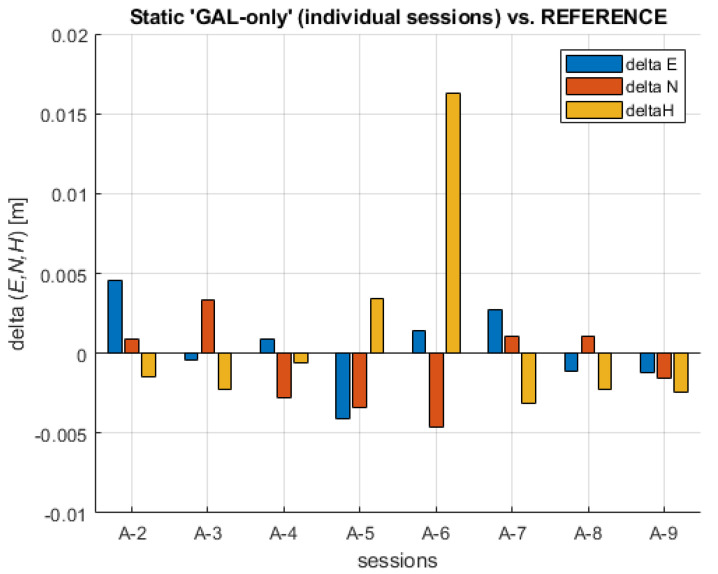
Coordinate differences (delta E, delta N, delta H) between individual static ‘GAL-only’ sessions and the REFERENCE.

**Figure 7 sensors-23-02466-f007:**
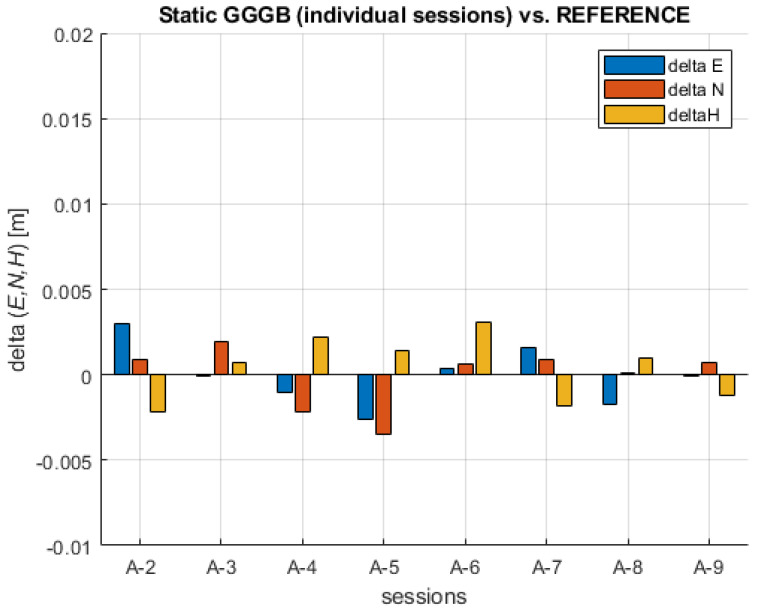
Coordinates’ differences between individual GGGB static sessions and the REFERENCE.

**Figure 8 sensors-23-02466-f008:**
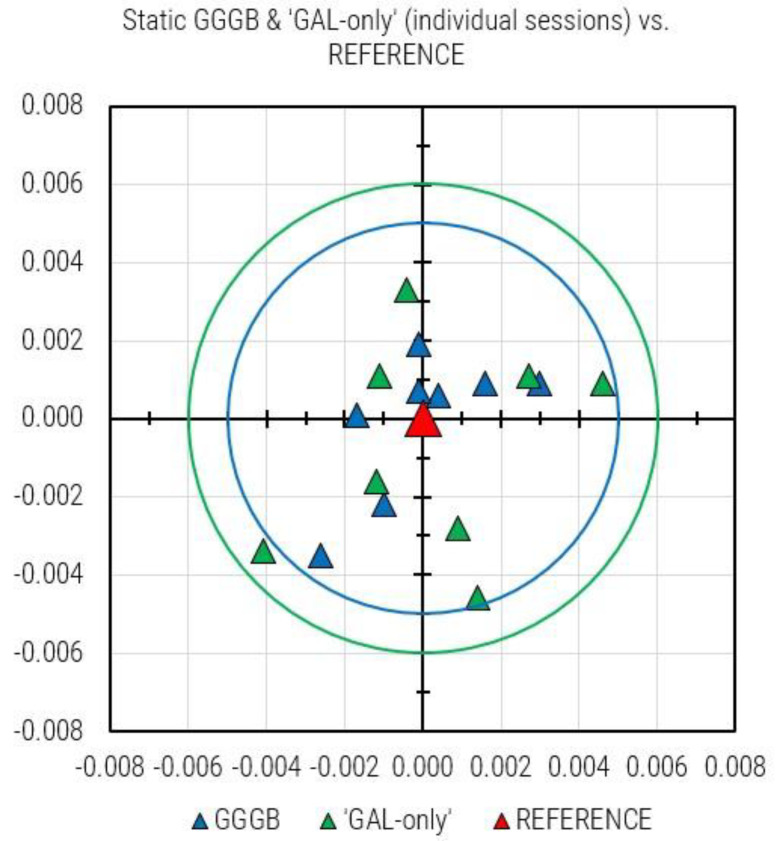
All individual static GGGB and ‘GAL-only’ sessions results plotted together: green circle (6 mm) and cyan circle (5 cm) show the maximum departure of ‘GAL-only’ and GGGB from the REFERENCE solution, respectively.

**Figure 9 sensors-23-02466-f009:**
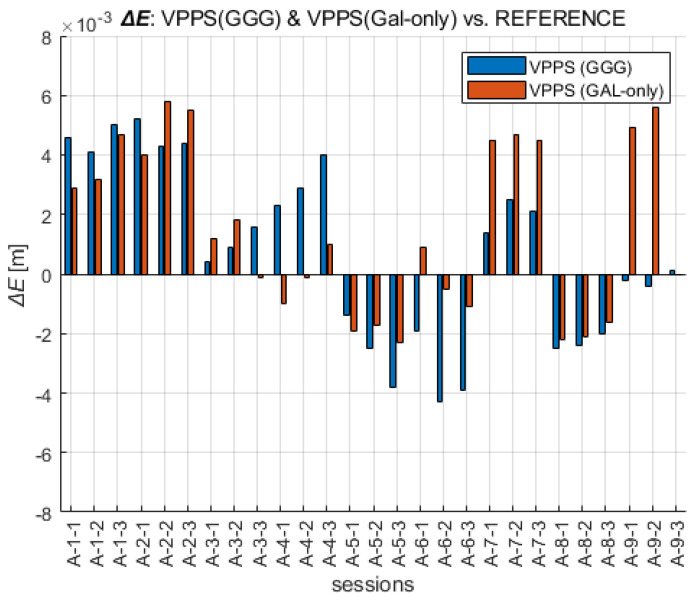
Easting coordinates obtained with VPPS using GGG (cyan) and GAL-only (green) observations given as difference to the REFERENCE value.

**Figure 10 sensors-23-02466-f010:**
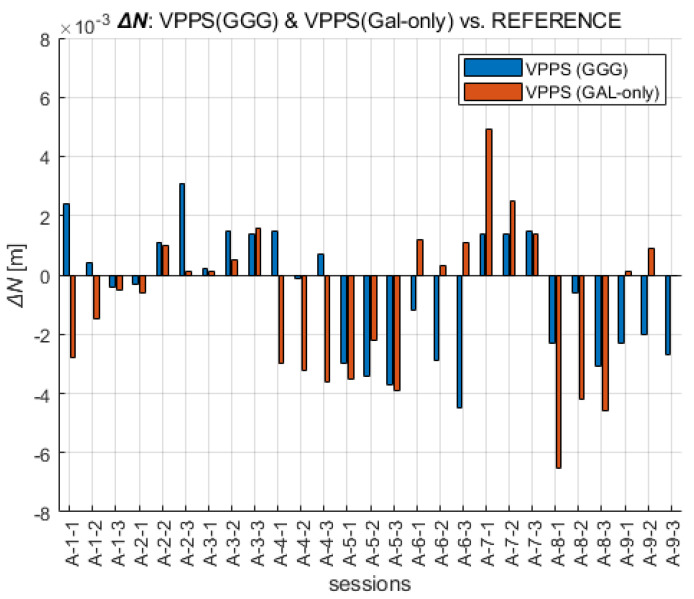
Northing coordinates obtained with VPPS using GGG (cyan) and GAL-only (green) observations given as difference to the REFERENCE.

**Figure 11 sensors-23-02466-f011:**
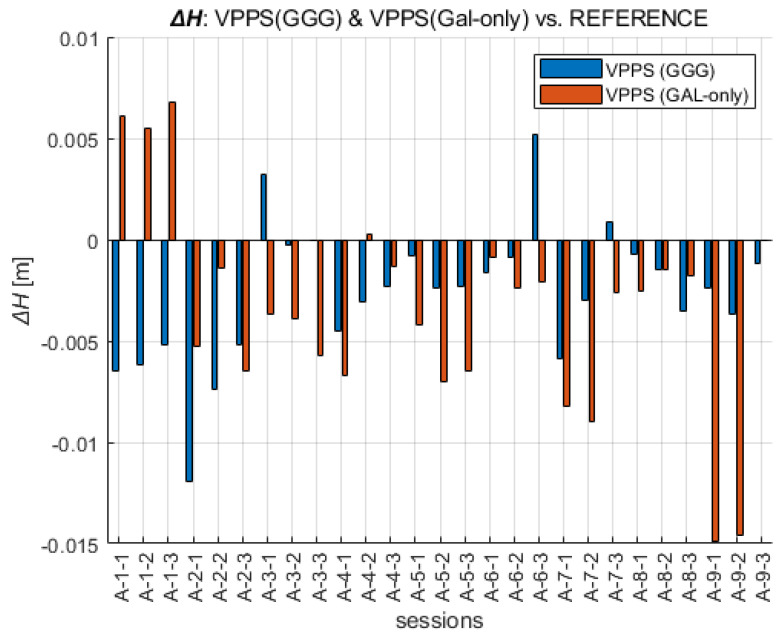
Height obtained with VPPS using GGG (cyan) and GAL-only (green) observations given as difference to the REFERENCE.

**Figure 12 sensors-23-02466-f012:**
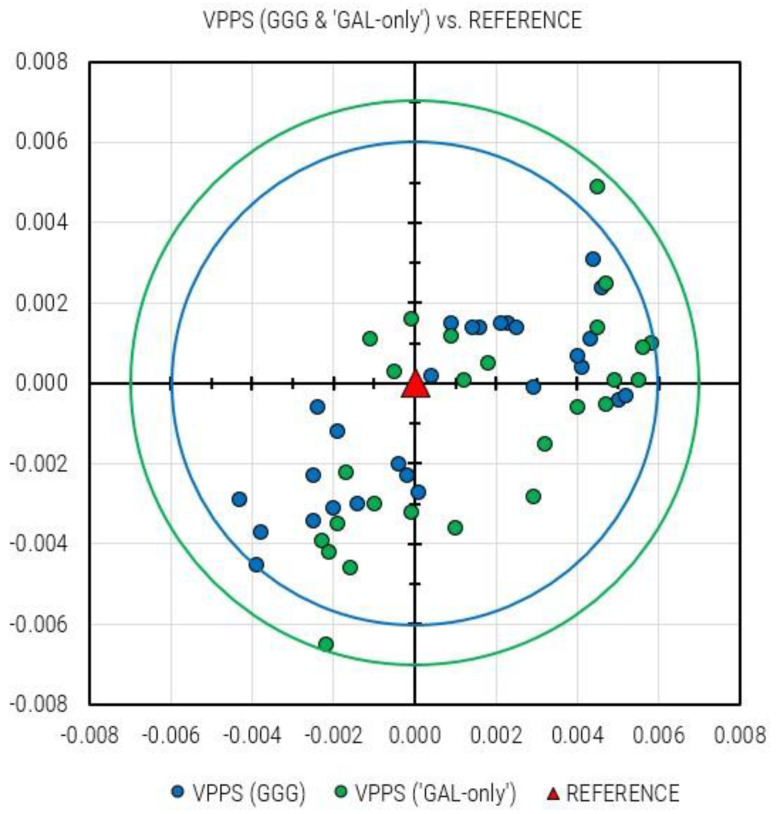
All VPPS solutions obtained: GGG (27 solutions; cyan dots) and GAL-only (26 solutions; green dots); the REFERENCE solution is marked with a red triangle.

**Figure 13 sensors-23-02466-f013:**
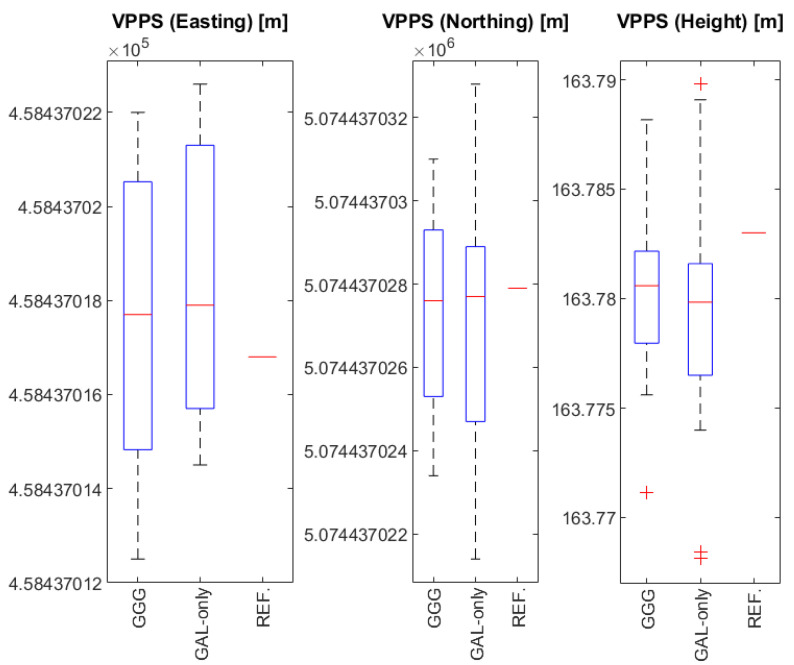
Box and whisker charts for Easting (**left**), Northing (**middle**), and Height (**right**) results obtained with VPPS (GGG), VPPS (GAL-only), and static GGGB daily solution (REF).

**Table 1 sensors-23-02466-t001:** Sessions planned according to the visibility of GAL satellites and expected PDOP values. Those having good, medium, and bad observation conditions are marked in green, orange, and yellow color, respectively.

Session Planned	Start of Session (CET)	Expected Number of GAL SV Visible	Expected PDOP
1	07:50	7	1.78
2	08:10	6	2.98
3	09:30	6	2.37
4	10:30	7	1.67
5	11:50	6	2.42
6	12:50	7	2.73
7	13:30	8	1.79
8	14:20	6	3.02
9	16:00	7	1.74
10	16:30	4	4.29

**Table 2 sensors-23-02466-t002:** Observation sessions and series of VPPS and GPPS (STATIC) as they were carried out.

Session	Series	Start Time (CET)	PDOP	Number of SV
1. (A-1)	VPPS (GGG)	07:53:08	1.2	18
	VPPS (GGG)	07:53:42	1.2	18
	VPPS (GGG)	07:54:18	1.2	18
	VPPS (GAL-only)	07:56:53	3.9	5
	VPPS (GAL-only)	07:58:38	3.9	5
	VPPS (GAL-only)	07:59:24	3.8	5
	STATIC (GGGB)	---	---	---
2. (A-2)	VPPS (GGG)	08:10:54	1.6	15
	VPPS (GGG)	08:12:06	1.6	15
	VPPS (GGG)	08:12:43	1.6	17
	VPPS (GAL-only)	08:20:50	3.3	5
	VPPS (GAL-only)	08:21:34	3.3	5
	VPPS (GAL-only)	08:22:10	3.3	5
	STATIC (GGGB)	08:25–08:47		
3. (A-3)	VPPS (GGG)	09:31:01	1.4	16
	VPPS (GGG)	09:31:47	1.4	16
	VPPS (GGG)	09:32:24	1.4	16
	VPPS (GAL-only)	09:34:55	2.4	6
	VPPS (GAL-only)	09:35:32	2.4	6
	VPPS (GAL-only)	09:36:11	2.4	6
	STATIC (GGGB)	09:38–10:00		
4. (A-4)	VPPS (GGG)	10:31:14	1.3	18
	VPPS (GGG)	10:31:52	1.4	18
	VPPS (GGG)	10:32:33	1.4	18
	VPPS (GAL-only)	11:02:30	3.0	6
	VPPS (GAL-only)	11:03:06	2.4	6
	VPPS (GAL-only)	11:03:43	2.4	6
	STATIC (GGGB)	11:05–11:27		
5. (A-5)	VPPS (GGG)	11:53:50	1.3	20
	VPPS (GGG)	11:54:25	1.4	19
	VPPS (GGG)	11:55:00	1.4	19
	VPPS (GAL-only)	11:56:50	2.4	6
	VPPS (GAL-only)	11:57:26	2.4	6
	VPPS (GAL-only)	11:58:01	2.4	6
	STATIC (GGGB)	12:01–12:23		
6. (A-6)	VPPS (GGG)	12:54:27	1.3	18
	VPPS (GGG)	12:55:09	1.3	18
	VPPS (GGG)	12:55:43	1.3	18
	VPPS (GAL-only)	12:59:33	2.7	7
	VPPS (GAL-only)	13:00:07	2.7	7
	VPPS (GAL-only)	13:00:43	2.7	7
	STATIC (GGGB)	13:02–13:20		
7. (A-7)	VPPS (GGG)	13:56:45	1.3	23
	VPPS (GGG)	13:57:25	1.3	23
	VPPS (GGG)	13:58:01	1.3	23
	VPPS (GAL-only)	13:59:39	2.4	6
	VPPS (GAL-only)	14:00:12	2.4	6
	VPPS (GAL-only)	14:00:54	2.4	6
	STATIC (GGGB)	13:30–13:52		
8. (A-8)	VPPS (GGG)	14:26:50	1.6	23
	VPPS (GGG)	14:27:30	1.3	23
	VPPS (GGG)	14:28:06	1.3	23
	VPPS (GAL-only)	14:30:22	3.0	6
	VPPS (GAL-only)	14:30:56	3.0	6
	VPPS (GAL-only)	14:31:33	3.0	6
	STATIC (GGGB)	14:32–14:55		
9. (A-9)	VPPS (GGG)	16:13:59	1.1	25
	VPPS (GGG)	16:14:32	1.0	26
	VPPS (GGG)	16:15:08	1.0	26
	VPPS (GAL-only)	16:17:08	1.6	7
	VPPS (GAL-only)	16:17:43	2.1	6
	VPPS (GAL-only)	----	----	----
	STATIC (GGGB)	16:31–16:53		

**Table 3 sensors-23-02466-t003:** Range of #SAT, PDOP, Hz Precision, and Vt Precision estimation values for solutions based on ‘GGG’ and ‘GAL-only’ satellites.

Solution		#SAT	PDOP	Hz Precision	Vt Precision
GGG	MAX	26	1.620	0.0043	0.0112
	MIN	15	1.012	0.0033	0.0050
GAL-only	MAX	7	3.910	0.0124	0.0218
	MIN	5	1.602	0.0044	0.0062

**Table 4 sensors-23-02466-t004:** Comparison of the number of satellites contained in the RINEX observation files (static) with number of satellites included in VPPS solutions (GGG and GAL-only).

Session	Number of SV (RINEX): GPS + GLO + GAL	Time (CET)	Number of SV (RINEX): GAL	Number of SV: VPPS (GGG)	Time (CET)	Number of SV: VPPS (GAL-only)	Time (CET)
2.	6 + 4 + 6 = 16	08:25	6	17	08:12	5	08:22
3.	8 + 5 + 6 = 19	09:38	6	16	09:32	6	09:36
4.	8 + 6 + 6 = 20	11:05	6	18	10:32	6	11:03
5.	8 + 5 + 6 = 19	12:01	6	19	11:55	6	11:58
6.	9 + 5 + 8 = 22	13:02	8	18	12:55	7	13:00
7.	9 + 7 + 6 = 22	13:52	6	23	13:58	6	14:00
8.	8 + 6 + 7 = 21	14:32	7	23	14:28	6	14:31

**Table 5 sensors-23-02466-t005:** Range values of solutions obtained with VPPS and GPPS.

Solution	Range (*E*)	Range (*N*)	Range (*H*)
VPPS (GGG)	9.5 mm	7.6 mm	17.1 mm
VPPS (GAL-only)	8.1 mm	11.4 mm	21.7 mm
GPPS (GGGB)	5.6 mm	5.4 mm	5.3 mm
GPPS (GAL-only)	8.7 mm	7.9 mm	19.4 mm

## Data Availability

Not applicable.

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
