# Peer review of "Assessment of GNSS Galileo Contribution to the Modernization of CROPOS’s Services"

_sensors, 2023, doi:10.3390/s23052466_

Round 1
Reviewer 1 Report
The paper presents an analysis of differential positioning using Galileo only and combinations of all constellations. The results are as expected and do not present any innovation. The paper has the merit of confirming the expected increase of reliability of the inclusion of Galileo.
At this level of very high positional accuracy, of few millimeters, the data analysis should include more decimal places. For example, in figure 8, the rounding to millimeter is clearly seen, hiding the actual spatial distribution of errors. Rounding to millimeters can also be found in other graphical presentations. For sure that the processing software can output latitude and longitude with 9 decimal places, which corresponds to approximately 0.1 mm precision. In the conversion to projected coordinates in meters, 4 decimal places should be kept in order to preserve the actual dispersion of errors.
Reviewer 2 Report
I have unfortunately decided to not recommend this article for publication. The first principal reason for my decision is that this study is lack of novelty.
Point 1: Line 145-146, I suggest to insert here the Galileo of each individual the Frequency/MHz and Observation Codes, in order to give more thoroughness to the topic. See (but is not mandatory): DOI 10.1007/s00190-008-0300-3, DOI 10.1007/PL00012776, DOI 10.1016/j.asr.2010.02.001, DOI 10.1007/978-981-13-7759-4_7, https://doi.org/10.3390/rs14163930.
Point 2: Line 322-333, In Figure 2, what caused the bad observation conditions.
Point 3: Line 516-517, In Figure 7, why is delta H bigger than delta N in session A-6 as same as Figure 6.
Point 4: Line 694-696, What do the blue dots in Figure 13 right represent? Besides, the figure 13 is not of sufficient resolution.
Point 5: The conclusion can be expressed in a few words and are listed separately.
Reviewer 3 Report
This paper evaluates the VPPS and GPPS services of the CROPOS network, which has the capability of tracking satellites of all running GNSS systems. In particular, the authors test the solution performance using GAL-only observations. I have one comment.
The results and conclusions are based on the receiver of Trimble, which is rather similiar to the reference stations of CROPOS network. As we know, the type of receiver and antenna in positioning may affect the solution accuracy in the precise coordinate estimation process.
Therefore, I would suggest authors to add more data to show VPPS and GPPS results using other brand of receivers/antennas, which is often the case in surveying that users may be equipped with various types of hardwares. With this modification, I believe the conclusions (especially accuracy) are more general.
Round 2
Reviewer 2 Report
The pictures and tables in the article are made by the Excel tools, please use professional software to draw again. Such as: MATLAB、Python, etc.
Author Response
Figures were drawn in MATLAB, an additional outlier has appeared, the analysis has been provided.

Reviewer 3 Report
Yes, I agree with you simutaneous obaservation is the best choice. But I would insist that a thorough test with different receiver/antenna brands other than TRIMBLE would be more important, as you are giving the service quantities of your network. Therefore I still recommend you to perform a few additional field experiments for a better representation.
Author Response
Additional measurements were carried out although we have not managed to provide a 'GPS agnostic' receiver capable of providing network RTK results based on 'GAL-only' observations. All that has been done, is addressed in the 'Response to Reviewer 3 Comments' file.
